# PRC2 promotes canalisation during endodermal differentiation

**Jurriaan Jochem Hölzenspies**[1], **Dipta Sengupta**[2,3], **Wendy Anne Bickmore**[2], **Joshua Mark Brickman**[1,4]*, **Robert Scott Illingworth**[2,3]*

1 Novo Nordisk Foundation Center for Stem Cell Medicine—reNEW, Faculty of Health and Medical Sciences, University of Copenhagen, Copenhagen, Denmark, 2 MRC Human Genetics Unit, Institute of Genetics and Cancer, University of Edinburgh, Edinburgh, United Kingdom, 3 Centre for Regenerative Medicine, Institute for Regeneration and Repair, The University of Edinburgh, Edinburgh, United Kingdom, 4 Department of Biomedical Sciences, Faculty of Health and Medical Sciences, University of Copenhagen, Copenhagen, Denmark

* joshua.brickman@sund.ku.dk (JMB); robert.illingworth@ed.ac.uk (RSI)

## Abstract

The genetic circuitry that encodes the developmental programme of mammals is regulated by transcription factors and chromatin modifiers. During early gestation, the three embryonic germ layers are established in a process termed gastrulation. The impact of deleterious mutations in chromatin modifiers such as the polycomb proteins manifests during gastrulation, leading to early developmental failure and lethality in mouse models. Embryonic stem cells have provided key insights into the molecular function of polycomb proteins, but it is impossible to fully appreciate the role of these epigenetic factors in development, or how development is perturbed due to their deficiency, in the steady-state. To address this, we have employed a tractable embryonic stem cell differentiation system to model primitive streak formation and early gastrulation. Using this approach, we find that loss of the repressive polycomb mark H3K27me3 is delayed relative to transcriptional activation, indicating a subordinate rather than instructive role in gene repression. Despite this, chemical inhibition of polycomb enhanced endodermal differentiation efficiency, but did so at the cost of lineage fidelity. These findings highlight the importance of the polycomb system in stabilising the developmental transcriptional response and, in so doing, in shoring up cellular specification.

## Author summary

Embryogenesis requires a tightly coordinated programme of cellular expansion and specialisation to gives rise to all of the cell- and tissue-types of the developing organism. In mammals this is controlled, in part, by chromatin modifications that help to establish the underlying gene expression patterns. However, limited numbers of cells make it hard to study these mechanisms directly in embryos. To overcome this, we employed a defined cell-differentiation system to experimentally model early embryonic development. Using this, in combination with pharmacological and genetic approaches, we identified DNA sequences with the potential to act as control switches at developmentally regulated genes. Surprisingly, we did not detect any loss of the polycomb-associated chromatin mark H3K27me3, despite evident transcriptional activation within the pool of cells. However,

**Data Availability Statement:** High level processed data is included as supplemental information. All raw microarray and next generation sequence (NGS) data is deposited in GEO under the following series accession (GSE286195). Computer codes

and the associated processing files for this study are available via GitHub (https://github.com/) in the 'Holzenspies_2025' repository.

**Funding:** This work was supported by a joint grant between JMB and WAB from the BBSRC (BBSRC_BB/H005978/1 (JMB) and BBSRC_BB/H008500/1 (WAB)). Work in the R.S.I. laboratory was supported by an MRC Career Development Award (MR/S007644/1) and Simons Initiative for the Developing Brain (SFARI - 529085). Work in the W.A.B. laboratory was supported by an MRC University Unit grant MC_UU_00007/2. Work in the Brickman laboratory was supported by Lundbeck Foundation (R198-2015-412, R370-2021-617 and R400-2022-769), Independent Research Fund Denmark (DFF-8020-00100B, DFF-0134-00022B, and DFF-2034-00025B), the Novo Nordisk Foundation (NNF21OC0070898 and 107012), the Danish National Research Foundation (DNRF116), and European Union (ERC, SENCE, 101097979). The Novo Nordisk Foundation Center for Stem Cell Medicine (reNEW) is supported by the Novo Nordisk Foundation (grant number NNF21CC0073729, and previously NNF17CC0027852). The funders had no role in study design, data collection and analysis, decision to publish, or preparation of the manuscript. DS was supported by a Royal Society-SERB Newton International Fellowship from the Newton-Bhabha Fund.

**Competing interests:** The authors have declared that no competing interests exist.

blocking deposition of this mark enhanced differentiation, but also led to reduced developmental fidelity due to increased production of inappropriate cell types. This demonstrated that, although acting as a secondary layer of gene regulation, polycomb marks are essential to stabilise developmental gene expression patterns and shore up cellular specification.

## Introduction

All vertebrates undergo a complex morphogenetic process known as gastrulation, in which embryonic differentiation is orchestrated in the midst of tissue rearrangements. As a result of this complex set of cell movements, the embryonic axes are specified and cells become exposed to rapidly changing signaling regimes. In mammals, cells will migrate through a transient embryonic region rich in signaling, the primitive streak (PS), and emerge as endoderm and mesoderm. This fast-moving phase of development is controlled by the convergence of these signalling pathways on a network of transcription factors (TFs). However, the dynamic nature of gastrulation and the number of diverse signaling pathways that require rapid response, necessitates an additional layer of regulation at the level of chromatin to restrict or promote transcription. This 'epigenetic' information operates as a molecular rheostat to ensure the appropriate response to developmental cues in order that the rapid inductive events occurring at this stage produce the precise set of cell types required for later development.

Polycomb group (PcG) proteins are paradigmatic developmental epigenetic repressors. PcGs combine to form functionally distinct complexes; principle amongst which are polycomb repressive complex one and two (PRC1 and 2). PRC1 can ubiquitylate chromatin on histone H2A on lysine 119 (H2AK119ub) through the action of its E3-ubiquitin ligase, RING1A or B [1,2]. Canonical PRC1 (cPRC1) also contains CBX and PHC subunits which bestow the capacity to shape chromatin architecture both locally and at a genome-wide scale [3–10]. PRC2 methylates histone H3 on lysine 27 (H3K27me1/2/3) via its histone methyltransferase (HMTase) subunit EZH1 or 2 [11–14]. The two PRCs have reciprocal affinity for the histone mark deposited by the other, and so co-operate to reinforce their recruitment onto chromatin [11,15–19].

Mutations that disrupt either PRC1 or 2 lead to developmental failure between implantation and gastrulation. Loss of RING1B (PRC1) leads to variable impairment and delayed development between embryonic days E6.5 –E8.5 in the mouse, with no embryos reaching the WT equivalent of the headfold stage (~E7.75) [20]. Embryos lacking either EED or EZH2 (PRC2) arrest at peri-implantation or during gastrulation [21,22]. In each case, abnormal anterior-posterior patterning is evident across the PS, with excess accumulation of presumptive extraembryonic mesoderm at the posterior end and a reciprocal depletion of embryonic mesoderm at the anterior [20–24]. Single cell RNA-seq (scRNA-seq) analysis of embryos deficient for EED identified enhanced production of posterior structures at the expense of those at the anterior end of the PS [25].

In mammals, gastrulation begins following the maturation of the epiblast from the inner cell mass (ICM) of the blastocyst [26–29]. Ex vivo culture of the ICM can be used to produce immortal cell lines with the capacity to differentiate into all the future lineages of the embryo and are therefore referred to as pluripotent. These cell lines can be differentiated, in the absence of morphogenetic movements and in a defined platform through a primitive streak like stage to generate definitive and anterior definitive endoderm (DE and ADE respectively). This model circumvents the small cell numbers (~15,000 cells by the end of gastrulation) and

highly heterogeneous nature of embryonic differentiation, to enable the purification of relatively homogeneous populations representing different stages of embryonic differentiation on adherent substrates. [30–32] (**Fig 1A and 1B**). We leverage this system in combination with genome-wide chromatin and transcriptional profiling and chemical-genetic approaches to investigate the interplay and dynamics of chromatin-based gene regulation during the epiblast to mesendoderm transition. While levels of the 'active' mark H3K4me3 are tightly correlated with transcriptional changes during differentiation, changes in levels of PRC2-mediated H3K27me3 are delayed relative to transcription. Acute inhibition of PRC2 activity enhanced endodermal differentiation but also resulted in the expression of lineage inappropriate genes. These findings highlight the importance of the polycomb system in stabilising appropriate transcriptional responses during embryogenesis and supporting Conrad Waddington's valleys [33]; the molecular canalisation that guides directional lineage specification.

## Results

### ESC differentiation models mouse primitive streak formation

To investigate the connection between the chromatin landscape and gene expression during early embryonic development, we adopted a monolayer differentiation strategy to convert mESCs into Anterior Definitive Endoderm (**Fig 1A and 1B** [30–32,34]). In this approach mESCs are differentiated through developmentally appropriate intermediates controlled by a temporally organised regimen of cytokines and growth factors (**Fig 1B**; ICM, epiblast, primitive streak, a transient endodermal population that expresses mesoderm and endoderm markers (mesendoderm) and ADE). To monitor this differentiation, we utilised a dual reporter mESC line (B6), which carries fluorescent reporters knocked in to the *Gsc* (GFP) and *Hhex* (Redstar) loci (**S1A Fig**; [31,35]). GSC is a marker of the PS/mesendoderm and HHEX marks definitive endoderm lineages. Together these reporters allow the quantitation and isolation of defined cell populations during differentiation.

Imaging and fluorescence-activated cell sorting (FACS) analysis monitoring expression of the fluorescent reporters showed the emergence of cells that start to express GSC on day 3 (d3) and this population grew rapidly and expressed increasingly high levels of GSC into d4. On d5, a GSC and HHEX double positive population emerged from the cells expressing the highest levels of GSC and this population continued to grow until d6 (**Figs 1B and S1B**). Immunostaining showed robust co-expression of CXCR4, a cell-surface marker of definitive mesendoderm [36,37], in GSC$^+$ cells throughout the differentiation (**S1B Fig**). These analyses suggest that cells undergoing differentiation progressed through stages equivalent to those observed in the embryo and converted to endoderm and the ADE with high efficiency.

Using FACS, we isolated nine cell populations as cells transition towards ADE (G = GSC; H = HHEX): d3 GSC negative or positive (G$^-$, G$^+$), d4 G$^-$ or G$^+$, d5 G$^-$, d5G$^{low}$; d5G$^{high}$, d6 G$^+$H$^-$, d6G$^+$H$^+$ (**S1C Fig**). These represent six defined developmental states: epiblast-like (d3, d4 and d5 G$^-$ populations), early PS (d3 and d4 G$^+$), posterior PS (d5 G$^{low}$), anterior PS (d5 G$^{high}$), anterior mesendoderm (d6 G$^+$H$^-$) and ADE (d6 G$^+$H$^+$) (**Fig 1B**). Following isolation, we confirmed that each population was stable and not subject to substantial drift between FACS gates, even following extended sort periods (2–4 h; **S1C Fig**). We validated the isolated populations using quantitative real time PCR (qRT-PCR). *Gsc* expression closely aligned with that of the reporter construct. *Hhex* mRNA was detected in both d6 reporter-negative and -positive populations, and to a lesser extent in the d5 G$^{high}$H$^-$ population. This may reflect allelic bias where some cells express *Hhex* from the endogenous allele instead of the reporter tagged allele. Nonetheless, the H+ population shows higher levels of both *Hhex* and the endodermal marker *Cer1*, suggesting isolation based on the reporter still enriches for ADE, even if not all ADE is included in this population (**S1D Fig**).

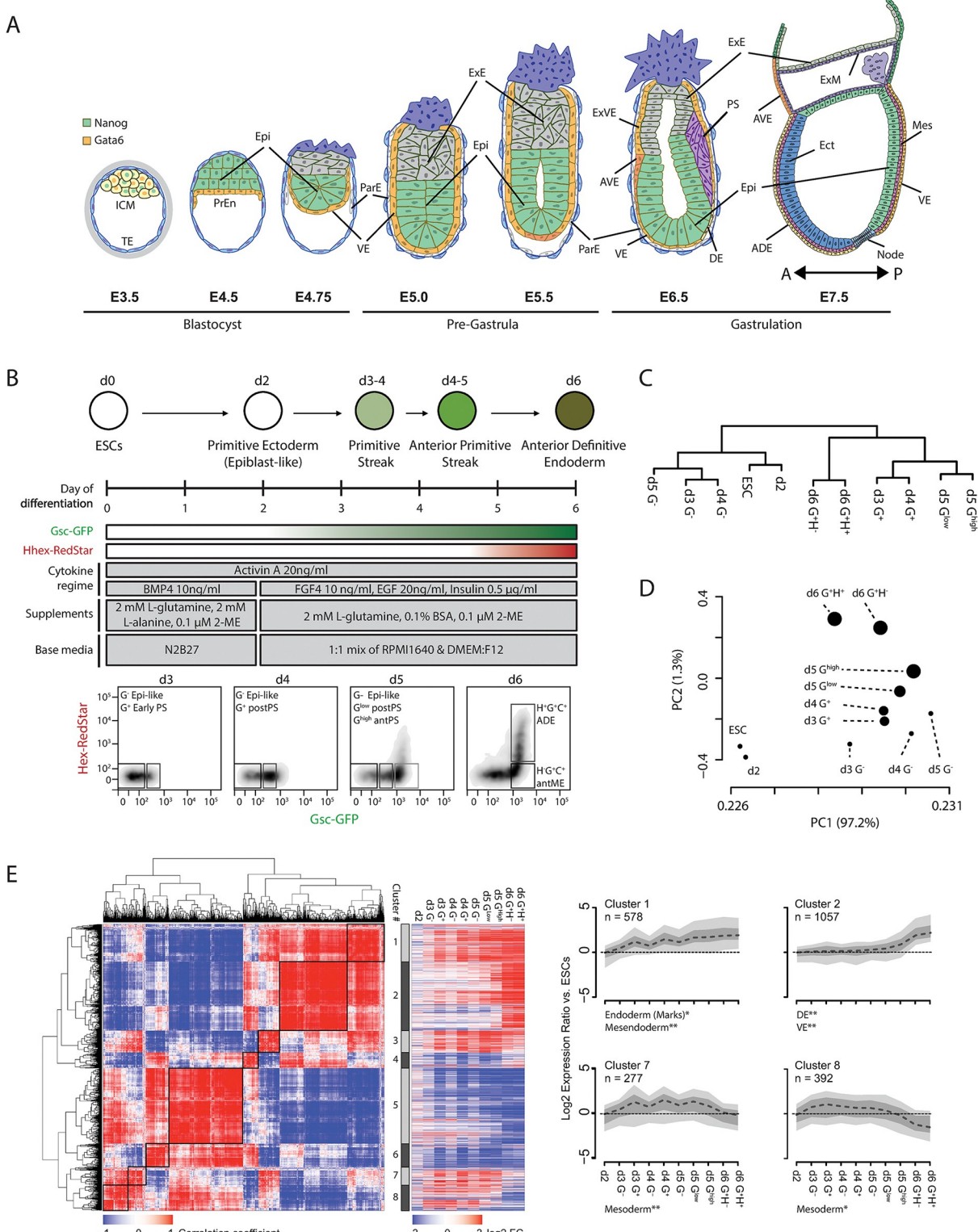

**Fig 1. Endoderm differentiation as an in vitro model of mouse gastrulation. A)** Schematic depicting murine blastocyst to gastrula stage embryonic development. Mutually exclusive expression of the pluripotency factor NANOG and the endoderm marker GATA6 are coloured as shown (top left). A-P denotes the anterior—posterior orientation of the embryo. Embryonic days (E) are shown relative to the gross developmental stage. Epi—Epiblast, PrEn—Primitive Endoderm, VE—Visceral Endoderm, ExE—Extraembryonic Ectoderm, ExVE—Extraembryonic Visceral Endoderm, AVE—Anterior Visceral Endoderm, DE—Definitive Endoderm, PS—Primitive Streak, ADE—Anterior

Definitive Endoderm, Mes–Mesoderm and Ect–Ectoderm. **B**) Differentiation scheme indicating the in vivo developmental counterparts at different days of ADE relative to GSC and HHEX reporter expression (upper panel). The media, supplement and cytokine regimen utilised throughout the ADE differentiation (middle panel). FACS profiles and gating based on the expression of the GSC-GFP (x-axis) and HHEX-RedStar (y-axis) at ADE d3-6 (lower panel; in vivo developmental counterparts of each sorted population noted). Epi-like—epiblast-like, Early PS—early primitive streak, postPS–posterior primitive streak, antPS–anterior primitive streak, antME–anterior mesendoderm and ADE–anterior definitive endoderm. **C**) Dendrogram depicting the hierarchical relationship between ESCs and ADE populations based on gene expression clustering (includes ESCs, unsorted d2 ADE and the FACS sorted ADE populations gated as in **B**. **D**) Scatter plot showing principal components 1 and 2 of PCA analysis performed on the same gene expression data samples as in **C** (circle size represents the level of GSC expression based on the FACS gating shown in **B**. The percentage variance attributed to each principal component (PC) is shown in the axis labels. **E**) Left; heatmap depicting the pairwise correlation scores for all genes differentially expressed during ADE differentiation (n = 3878 RefSeq genes). Genes are arranged based on hierarchical clustering as denoted by the dendrogram plots. Boxed regions indicate the eight clustered gene subsets that show similar expression profiles during the differentiation. Middle; heatmap showing gene expression changes ($\log_2$ ADE/ESC) corresponding to the eight gene clusters. Right; aggregate gene expression profile of the genes in clusters 1, 2, 7 and 8. The heavy dashed line, dark grey shaded area and light grey shaded areas represent the median, 25th to 75th percentile range and 10th to 90th percentile range of the log2 fold change for each gene set respectively. Significantly enriched functional gene sets for each of the clusters are indicated below their respective plot (p≤0.05* and p≤0.01** using a Fischer's exact test with Benjamini & Hochberg multiple testing correction; **S2 Table**).

In contrast, *Pou5f1*, encoding the pluripotency TF OCT4, showed reduced expression concordant with the time and extent of differentiation. Thus, candidate analysis indicates that the *in vitro* cell populations resemble their *in vivo* counterparts.

We performed genome-wide expression analysis across mESCs, day 2 (unsorted) and the nine cell populations described above. Linear modelling and statistical analysis identified extensive differential gene expression ($\log_2$ FC $\geq$ 1.5 and a p value of $\leq$ 0.01) between all these populations, the magnitude and extent of which scaled with the extent of differentiation (**S2A Fig** and **S1 Table**). Hierarchical clustering based on all normalised gene expression values grouped the populations primarily based on the level of GSC and HHEX expression rather than days under differentiation. Four clear groups emerged including d2/ESCs, d3-5 G⁻, d3-5 G^{low/high/+} and d6 G⁺H^{+/-} corresponding to cells of the ICM, epiblast, PS and mesendoderm/ ADE respectively (**Fig 1C**). Principal Component Analysis (PCA) yielded equivalent group-ings, but with a more evident contribution of time-in-culture (**Fig 1D**). Expression of key marker genes changed with the expected dynamics during differentiation (**S1D Fig**). To per-form a more unbiased assessment of the *in vivo* relevance of ESC to ADE conversion, we first identified all genes that showed differential expression during the differentiation (n = 3,878 Refseq genes; **S1 Table**). Un-supervised clustering based on pairwise correlation scores between the $\log_2$ fold change (FC) expression profiles of each of these genes across the ten ADE populations (i.e., $\log_2$^{ADE sample X/ESC}) identified eight distinct clusters (**Fig 1E** and **S1 Table**). For seven of the clusters, gene set analysis on gene lists (**S2 Table**) yielded significant enrichment of defined developmental terms. For example, cluster 1 genes displayed gradual but consistent upregulation and are enriched for endodermal and mesendodermal markers, whereas cluster 2 genes are upregulated later (largely restricted to the d6 populations) and enriched for markers of the DE and VE (**Fig 1E** and **S1** and **S2 Tables**). Clusters 7 and 8, which are enriched for mesodermal genes, show a phased period of upregulation followed by down-regulation. Analysis on the regions surrounding the TSS of these genes (-100 bp to +50 bp) confirmed enrichment of TF motifs consistent with these developmental categories (**S2B Fig**). Enrichment for VE as well as DE genes in cluster 2 is expected given their developmental proximity and the fact that VE cells can contribute to DE (**Fig 1A**) [34].

## Transcriptional changes determine the gene expression landscape during the epiblast to primitive streak transition

Gene expression assays based on total cellular RNA reflect not only changes in transcription but also the sum of post-transcriptional processing, RNA stability and RNA turnover. To

understand the highly dynamic transcriptional changes associated with the beginning of gastrulation, we focused on the in vitro equivalent of epithelial to mesenchymal transition (EMT) at the PS stage and used 4sU-seq to examine nascent transcriptional changes in the G$^-$ and G$^+$ populations from d3 and 4. The 4sU-seq profiles display prominent intronic signal indicative of isolation of un-processed pre-mRNA transcripts and inspection of individual loci confirmed the expected transcriptional levels of candidate genes: *Gapdh* (ubiquitous), *Gata4* (endoderm), *Gsc* (mesendoderm) and *Pou5f1* (pluripotency; **Fig 2A**).

We quantified transcript levels across all genes and performed statistical analysis with equivalent parameters to those used for total RNA levels (**S3 Table**). Direct comparison of differential gene expression between nascent transcripts (4sU-seq) and total RNA levels (microarray) revealed that most gene expression changes identified in total RNA can be attributed to changes in transcription (**Fig 2B** and **S3 Table**). In line with this we noted that, whereas very few genes were differentially expressed between the d3 G$^-$ and G$^+$ populations in total RNA alone (3 upregulated and 27 downregulated genes), a larger set of genes was differentially expressed in the 4sU-seq dataset alone (**Fig 2B**; highlighted genes—lower left panel). One day later (d4), these genes now became significantly up- or down-regulated in the total RNA data (**Fig 2B**; highlighted genes—lower right panel).

Overall transcript abundance was not responsible for this observation. Comparing the 4sU-seq signal of differentially expressed genes for those identified in '*both*' datasets to those specific to the '*4sU*'-seq data alone, as the signal intensities between the two categories were similar ('*Both*' vs. '*4sU*'; **Fig 2C**). We noted that there was a significant difference in transcript levels between the d3 and d4 G$^-$ populations for all differentially expressed genes, but not for the respective G+ populations (**Fig 2C**). This was likely due to the smaller dynamic range of GSC expression in the G$^+$ vs. G$^-$ FACS gates allowing for a greater degree of population (and therefore gene expression) drift within these populations between d3 and d4 (**Fig 1C**). This was supported by the observation that d3 and d4 G$^+$ populations clustered more tightly in PCA analysis compared to their G$^-$ counterparts (**Fig 1D**).

Upregulated genes identified in 4sU-seq alone are significantly longer than those identified in both expression datasets, consistent with larger transcripts taking longer to process into mature mRNAs (**Fig 2D**) and, as expected, genes upregulated only in the 4sU-seq dataset showed a high proportion of unspliced transcripts (lower exon/intron ratio) than genes upregulated in both datasets (**S3A Fig**). These findings demonstrate we had the temporal resolution to separate the activation of transcription from the subsequent accumulation of processed mRNA 24h later in sorted cell populations from this differentiation model.

## Dynamic intergenic transcription signals identify putative enhancers in endoderm specification and patterning

Developmental gene expression programmes are established, in part, through the binding of tissue and cell-type TFs to enhancers. Active enhancers are marked by bi-directional transcription, producing short, unstable eRNAs that are detectable by 4sU-seq [38,39]. Using the dREG algorithm [40,41] we identified 24,862 'peaks' of bi-directional transcription in the d3 and d4 G$^+$ and G$^-$ 4sU-seq data and mapped them with respect to their neighbouring genes (**S4 Table**). Excluding peaks that overlap with, or are proximal to, known coding regions (within -3 kb to +20 kb of a gene TSS; **S4 Table**) left a subset of 4,127 intergenic loci. We then quantified the 4sU-seq signal for all intergenic dREG-peaks located within 100 kb of a gene differentially expressed between the d3 G$^+$ and G$^-$ populations. As a group, these prospective enhancers display differential transcription that was equivalent in direction to that of their neighbouring gene at $\leq$ 100 kb separation (n = 123 and 100 for down- and up-regulated genes

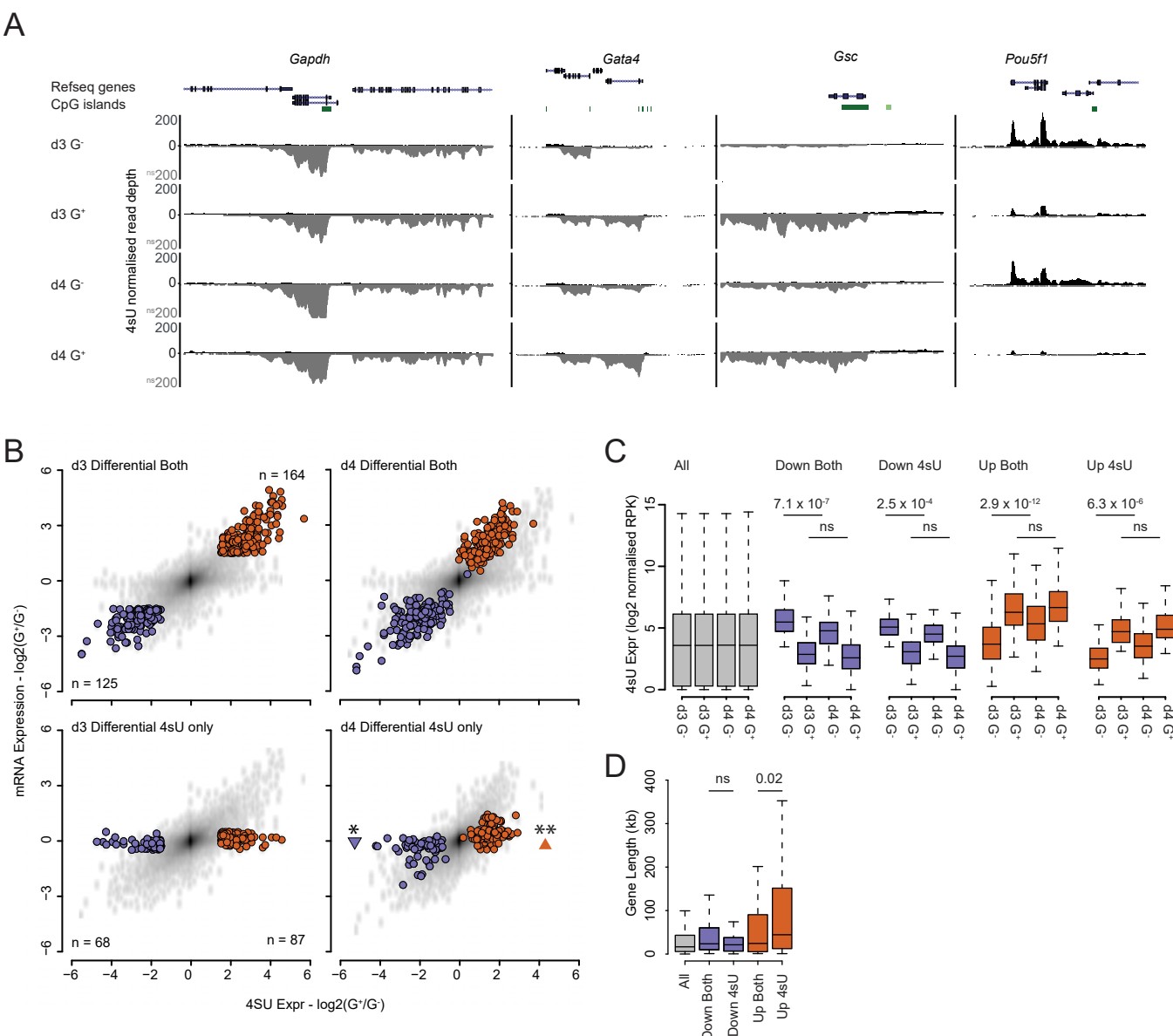

**Fig 2. Transcription is the dominant process governing mRNA levels in differentiation. A**) Genome browser tracks of normalised 4sU-seq signal at candidate gene loci (*Gapdh*, *Gata4*, *Gsc* and *Pou5f1*) for d3 and d4 ADE populations. The signal is coloured according to the transcribed strand (positive—black and negative—grey). The tracks represent a single matched experiment (1 of 3 independent replicates). Genes are annotated as per the mm9 genome assembly. **B**) Scatter plots comparing $\log_2$ fold expression changes ($\log_2$ G⁺/G⁻) between 4sU-seq signal (x-axis) and mRNA signal (y-axis) for d3 and d4 ADE populations. Genes which are significantly up- or down-regulated in d3 G⁺ vs. G⁻ populations are indicated in red and blue respectively. Significant changes in gene expression were determined using a Wilcoxon rank-sum test ($p \leq 0.05^*$ and $p \leq 0.01^{**}$). The number of RefSeq genes in each set are indicated. **C**) Boxplots showing the normalised reads per kb (RPK) values for gene body 4sU-seq signal for all (grey), upregulated (red) or downregulated (blue; in ADE d3 G⁺ vs. G⁻) genes for the indicated ADE populations. Significant changes in 4sU-seq signal were determined using a Wilcoxon rank-sum test (ns–non significant and p values as displayed). **D**) Boxplots showing gene length in kilobases for all (grey), upregulated (red) or downregulated (blue; in ADE d3 G⁺ vs. G⁻) genes for the indicated ADE populations. Significant changes in gene length were determined using a Wilcoxon rank-sum test (ns–non significant and p values as displayed). Data shown in (**C**) and (**D**) represent the mean of three independent replicate experiments.

respectively; **Fig 3A** and **S4 Table**). This transcriptional linkage was also observed at d4. Examination of transcriptional profiles at dREG-peaks showed a prominent bi-directional transcriptional signature whose magnitude correlated with the expression of the associated neighbouring (within 100 kb) gene (**Figs 3B–3D** and **S4A**). A consistent association between

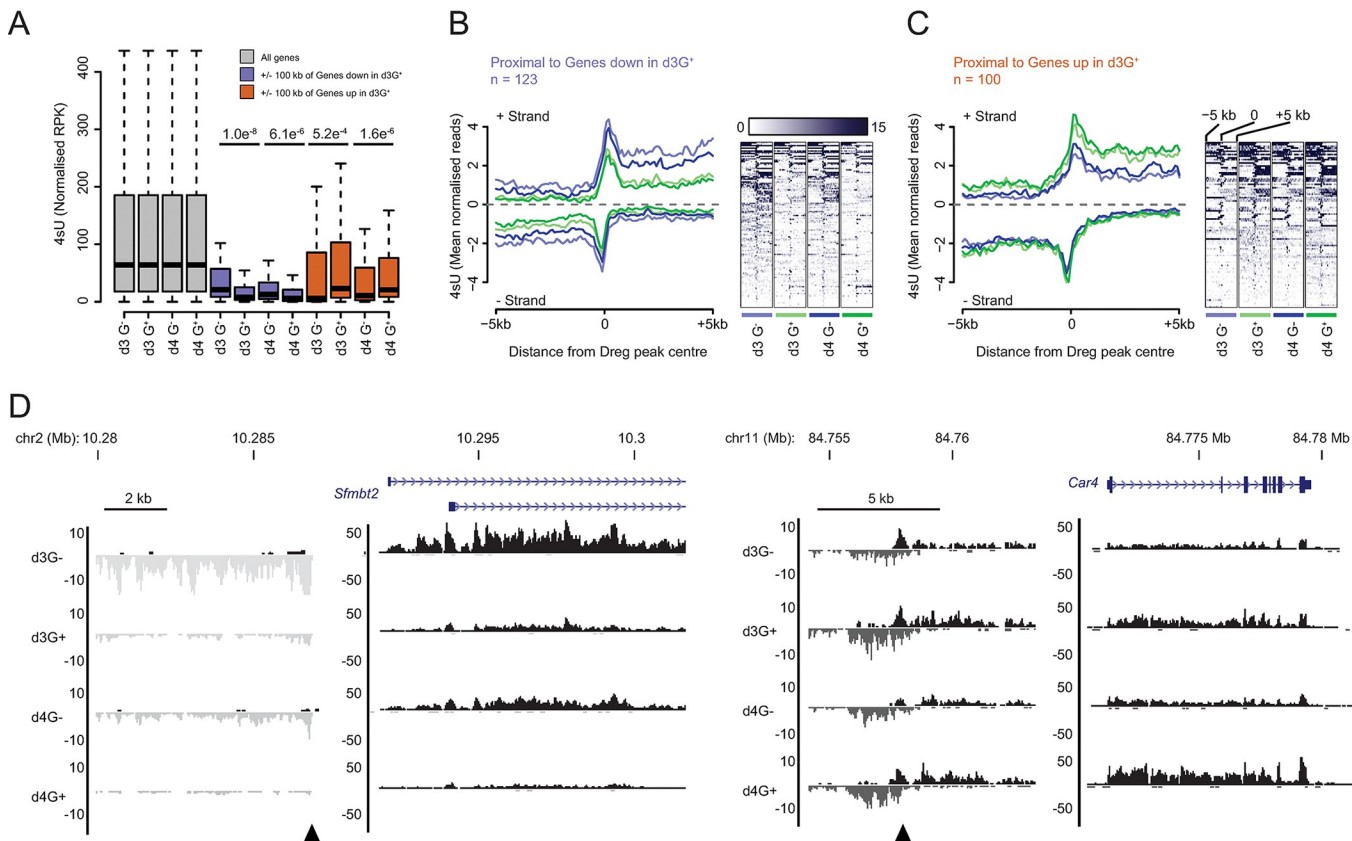

**Fig 3. Transcriptional profiling identifies prospective enhancers in ADE. A**) Boxplots of 4sU-seq signal (normalised reads per kilobase—RPK) at dREG peaks that are proximal to (+/- 100 kb) the TSS of all (grey), downregulated (d3 G⁺ vs. G⁻; blue) and upregulated (d3 G⁺ vs. G⁻; red) genes. Significant changes in 4sU-seq signal were determined using a Wilcoxon signed rank test (p values as displayed). **B** & **C**) 4sU-seq signal profiles and heatmaps of dREG peaks (+/- 5 kb) that are proximal to (**B**) downregulated and (**C**) upregulated genes. The numbers of genes are indicated ('n = ') and defined as in **A**. Heatmaps depict the 4sU-seq signal at each dREG peak (one peak/genomic interval per row) and show the combined 4sU-seq signal that maps to either DNA strand. Heatmaps are presented relative to the peak centre and oriented 5' to 3' according to the genome build and not relative to the transcriptional orientation of the closest/associated gene. The profile plots represent the mean 4sU-seq signal (mean profile for the regions shown in the heatmaps) with strand-specific transcripts shown above and below the x-axis (mapping to the + and–strand respectively). The mean profiles plots are colour-coded to match specific heatmaps (colour coding as indicated below teach heatmap). **D**. Genome browser tracks of normalised 4sU-seq signal at candidate differentially expressed gene loci (*Sfmtb2* and *Car4* on chromosomes 2 and 11 respectively) and their upstream proximal dREG peaks. Data and genes are presented as in **Fig 2A** and dREG peaks are indicated with black arrow heads. Data shown in (**A—D**) represent the mean of three independent replicate experiments.

the transcription status of dREG-peaks and their nearest neighbouring gene degraded at distances beyond 100 kb (**S4B Fig** and **S4 Table**), suggesting that enhancers active during these early stages of differentiation are quite close to their target genes.

## Loss of H3K27me3 is delayed relative to transcriptional upregulation

Whilst TF binding to genes and enhancers dictates developmental expression patterns, histone modifications are thought to 'fine-tune' this process by controlling how transcriptionally permissive the underlying chromatin is. Two key modifications that are thought to operate in this way are H3K4me3 and H3K27me3; marks associated with the COMPASS (MLL/SET1; activating) and PRC2 (repressive) complexes respectively [11–14,42]. These antagonistic modifications frequently co-localise at the promoters of developmental genes either on the same bivalent nucleosome—a chromatin state thought to poise genes for subsequent transcriptional activation or repression [43–48], or on different alleles and/or in different cells within a population (bistable) [49]. Given that mutations in either MLL/SET1 and PRC2 complexes lead to

gastrulation failure [20–24], we determined the dynamics of these histone modifications during ADE differentiation.

We performed native chromatin immunoprecipitation for both H3K4me3 and H3K27me3 followed by deep sequencing (ChIP-seq) in each of the 9 differentiating populations. Inspection of the ubiquitous expressed *Gapdh* locus showed prominent H3K4me3 enrichment over the promoter CpG island (CGI) and an absence of detectable H3K27me3 across all cell populations (**S5A Fig**). Though the silent *Hoxb* gene cluster showed domain-wide enrichment of H3K27me3 and modest levels of H3K4me3 primarily focussed on CGIs throughout differentiation (**S5B Fig**), levels of H3K4me3 decreased and H3K27me3 increased at latter stages of differentiation. Both *Gsc* and *Hhex* were decorated with H3K4me3 and H3K27me3 prior to their activation and then underwent a reciprocal gain of H3K4me3 and loss of H3K27me3 in the later phase of differentiation, with *Hhex* showing slower resolution of the marks correlating with its relatively delayed activation (**Fig 4A and 4B**). ChIP-seq profiles at the TSSs of genes categorised based on their expression dynamics during differentiation (**Fig 1E, S1 and S5 Tables**) identified a clear correlation between H3K4me3 levels and gene expression (**Fig 4C and 4D**). In contrast, H3K27me3 levels showed a pronounced anti-correlation with gene expression levels with prominent losses and de novo gains when genes were up- and down-regulated, respectively (**Fig 4C and 4D**).

This analysis necessarily focussed on the relatively small fraction of genes whose expression changed significantly during the course of the differentiation (~18% or 3,878 of the 22,040 Refseq genes analysed; **Fig 4C and 4D**). However, to determine if chromatin modification state could stratify all genes independently of gene expression, we performed the reciprocal analysis by clustering TSSs based on H3K4me3 and H3K27me3 levels. The resulting gene sets were distinct with respect to promoter sequence composition (+/- CGIs), gene expression dynamics and importantly, functional classification (**S5C–S5E Fig, S5** and **S6 Tables**). Whilst dynamic chromatin profiles identified gene sets with variable expression levels (e.g. clusters 1, 4 and 8); constitutively low/low (cluster 2), low/high (cluster 10), high/low (cluster 5) or high/high (cluster 9) levels of H3K4me3 and H3K27me3 enriched for genes associated with immune, neuroectodermal, housekeeping and broad-developmental functions respectively (**S5C–S5E Fig, S5** and **S6 Tables**). This demonstrated that H3K4me3 and H3K27me3 levels could effectively stratify genes independently of their expression status during ADE differentiation.

To gain a better insight regarding the contribution of H3K4me3 and H3K27me3 to transcriptional regulation we identified genes which were consistently differentially expressed between the G$^+$ and G$^-$ populations at d3, 4 and 5 of differentiation (n = 52 and 97 for up- and down-regulated genes respectively; **Fig 5A and 5B** and **S5 Table**). H3K4me3 levels surrounding the TSS of these genes showed a significant change concomitant with the changes in gene expression, at least at the temporal resolution of this differentiation (24 h; **Fig 5C and 5D** and **S5 Table**). H3K27me3 levels showed an anti-correlated dynamic with respect to gene expression, however significant loss of H3K27me3 at up-regulated genes was not observed until d4 and d5 of differentiation, genes upregulated in the d3 data showed no loss of this histone modification at this time-point (**Fig 5E and 5F** and **S5 Table**). To determine if this delay might be due to the dynamics of RNA processing and catabolism, we repeated the analysis comparing H3K27me3 levels at genes differentially transcribed between d3 and d4 populations using the 4sU-seq data (n = 132 and 251 for up and down-regulated genes respectively; **S6A and S6B Fig** and **S5 Table**). This confirmed that H3K27me3 levels at gene TSSs did not significantly change despite their transcriptional activation in the d3 G$^+$ population (**S6C–S6F Fig** and **S5 Table**).

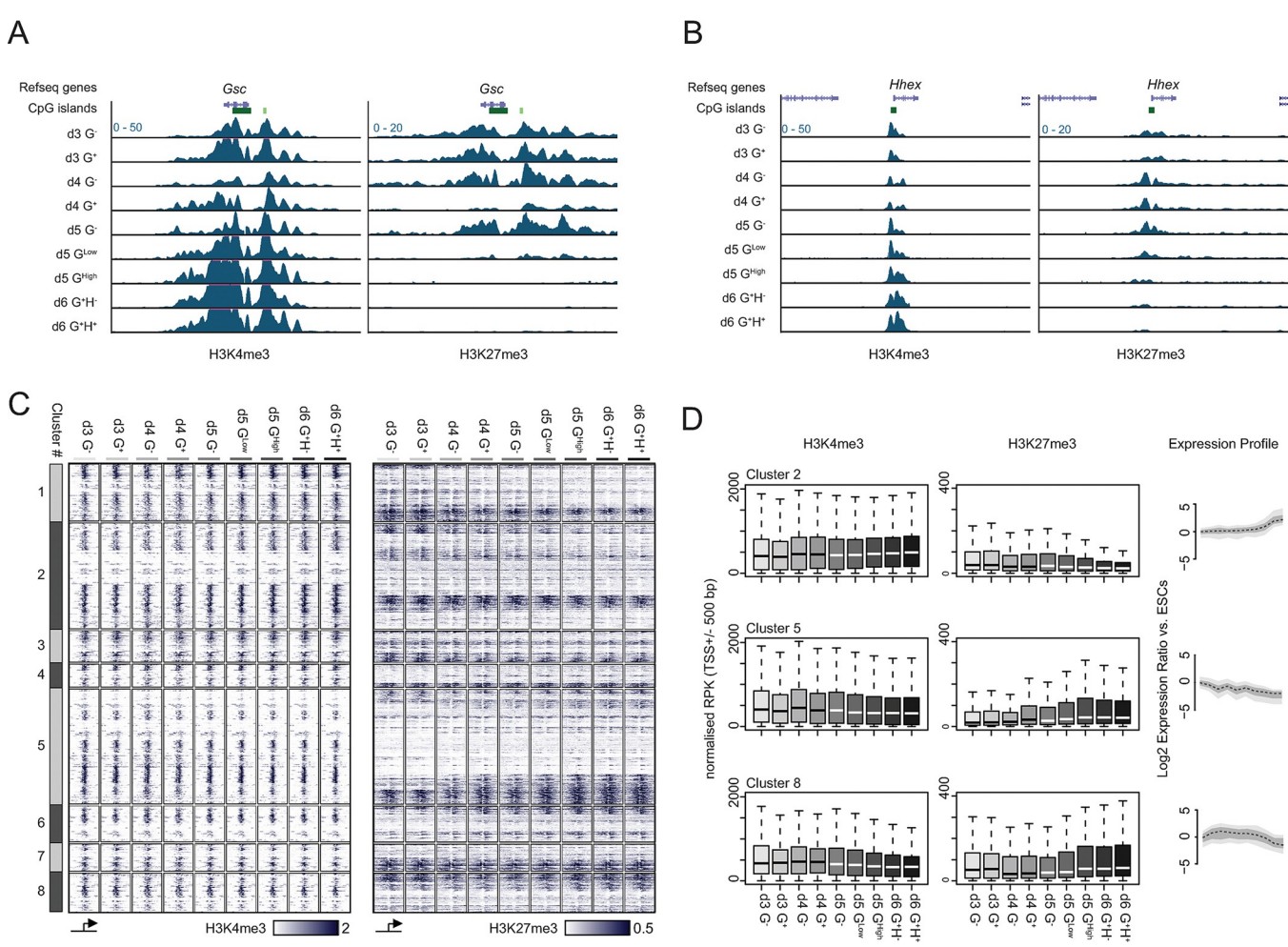

**Fig 4. Dynamic changes in H3K4me3 and H3K27me3 during differentiation. A** & **B**) Genome browser tracks of ChIP-seq signal for H3K4me3 (left panel) and H3K27me3 (right panel) at the *Gsc* (**A**) and *Hhex* (**B**) gene loci. Genes are annotated as per the mm9 genome assembly. **C**) Heatmaps of H3K4me3 (left) and H3K27me3 (right) ChIP-seq signal spanning +/- 5 kb of the TSS of differentially expressed genes. Heatmaps are sub-divided into the gene expression groups defined in **Fig 1E** and ordered by clustering TSS (+/- 500 bp) ChIP-seq signal from d3 G⁻ and d6 G⁺H⁺ populations for both histone modifications. The relative signal scale for each modification is shown below their respective heatmap. **D**) Summary boxplots of normalised H3K4me3 and H3K27me3 ChIP-seq signal at the TSSs (+/- 500 bp) for genes in clusters 2, 5 and 8 (left and middle panel respectively). The matched gene expression profile for these clusters is shown for reference (right panel; presented as in **Fig 1E**). Data shown in (**A—D**) represent the mean of two independent replicate experiments.

However, it is possible that H3K27me3 levels were invariant because i). activation occurred primarily at genes with little or no H3K27me3 at their TSS or ii). heterogeneous transcriptional activation from only a small fraction of the cells within the sorted population. Our FACs isolation strategy used to prepare cell populations for ChIP, microarray expression and 4sU-seq analysis was based on expression (or lack thereof) of the fluorescent GSC-GFP reporter (**Figs 1** and **S2A**). This means that every cell in each G⁺ population expressed the GSC fluorescent reporter (at least at the protein level) at the point of isolation. This level of induced transcription occurred despite the presence of appreciable and invariant levels of H3K27me3 at the TSS of the *Gsc* gene in both G⁺ and G⁻ populations at day 3 of differentiation (**S7A Fig**). H3K27me3 levels at the *Gsc* gene were then substantially reduced in the G⁺ relative to the G⁻ populations at day 4 comparable to what was observed for all upregulated genes (**Figs 5, S6** and **S7A**). As *Gsc* was transcribed in all G⁺ cells despite high levels of H3K27me3, this argued against low H3K27me3 or heterogenous expression as an explanation. However, it is possible

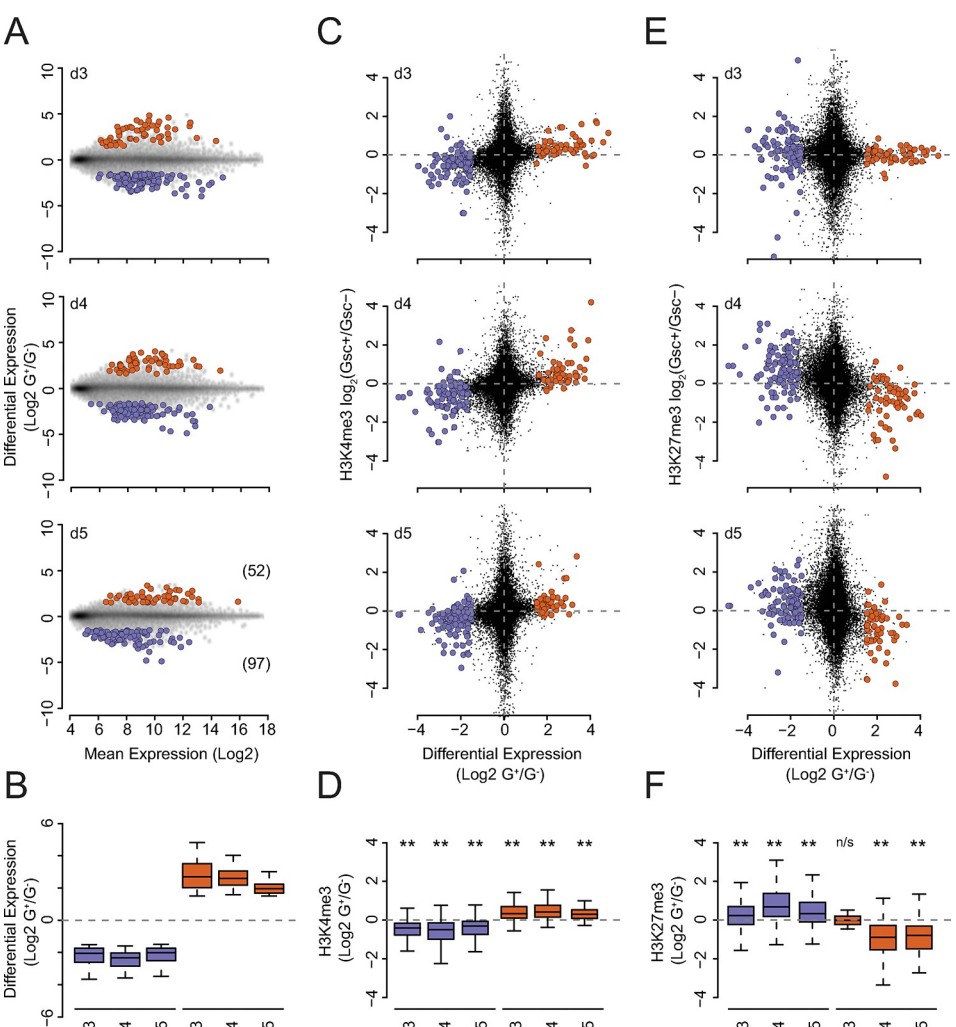

**Fig 5. Changes in TSS H3K27me3 levels are temporally subordinate to changes in gene expression. A**) MA plots of gene sets that show consistent differential expression between the $G^+$ and $G^-$ populations at d3, 4 and 5 of differentiation. Upregulated and downregulated genes are coloured in red and blue respectively and the number of genes are noted in parenthesis in the lower panel. **B**) Summary boxplots of $\log_2 G^+/G^-$ ratios for the gene and datasets shown in **A**. **C**) Scatter plots depicting the $\log_2 G^+/G^-$ expression ratios (x axis) vs. the $\log_2 G^+/G^-$ ratio of H3K4me3 ChIP-seq signal at the TSS (+/- 500 bp; y axis) of the differentially expressed genes shown in **A**. **D**) Summary boxplots of the $\log_2 G^+/G^-$ ratio of H3K4me3 ChIP-seq signal at the TSS (+/- 500 bp; y axis) for the gene and datasets shown in C. Significant changes between the populations for each day for the raw, unlogged values, are indicated above their respective plot (p≤0.05* and p≤0.01** determined using a Wilcoxon signed rank test). **E**) As for panel (**C**) but for H3K27me3 ChIP-seq data. **F**) As for (**D**) but for H3K27me3 ChIP-seq data. 'n/s' denotes a non-significant (>0.05) test result from a Wilcoxon signed rank test. Gene expression data shown in (**A** and **B**) and ChIP-seq data shown in (**C—F**) represent the mean of three and two independent replicate experiments respectively.

that *Gsc* represented an isolated example, and so we tested this more generally. We selected upregulated TSSs that had a greater than median value for both transcription (4sU-seq) and H3K27me3 in the d3 $G^+$ population (n = 35 of 125 genes; **S7B Fig**). Both combined and candidate-based analysis of these genes showed that transcriptional activation occurred in the presence of substantial and invariant levels of H3K27me3 at day 3 followed by a subsequent reduction by day 4 of differentiation (**S7B–S7D Fig**). Importantly, many of these genes yielded high absolute 4sU-seq signal, comparable to that observed for *Gsc* (**S7A Fig**), arguing against transcriptional activation being limited to only a small subpopulation of cells.

## PRC2 inhibition enhances anterior endoderm differentiation but at a cost to lineage fidelity

As it is a robust model of primitive streak formation, our differentiation model reflects the stage at which PRC dysfunction leads to embryonic failure [20,21,24]. To investigate the impact of perturbed polycomb function on ADE differentiation we blocked PRC2 catalytic activity during ADE differentiation using the potent EZH2 inhibitor EPZ6438 (henceforth referred to as EPZ) [50]. We performed three parallel treatment regimens in which: i). both mESCs culture and differentiation were carried in the presence of DMSO ('Control'), ii). mESCs were pre-treated with EPZ prior to differentiation in the presence of DMSO ('ES_EPZ') and iii). mESCs were pre-treated with DMSO prior to ADE differentiation in the presence of EPZ ('ADE_EPZ'). Immunoblotting of lysates taken before and during ADE differentiation showed a pronounced reduction in global H3K27me3 levels in the presence of EPZ demonstrating the efficacy and dynamics of the EPZ treatment in each treatment regimen (**Fig 6A**).

FACS analysis on d3—d6 populations differentiated under each of the three conditions (**Fig 6B and 6C**) showed a subtle delay in the expansion of the $G^+$ populations in d3 EPZ-pre-treated cells (ES_EPZ). However, this trend rapidly reversed with the expansion of the $GSC^{high}$ population relative to the control in d4 and d5. The ADE_EPZ differentiation was indistinguishable from controls until d5 where it too showed an expansion of the $GSC^{high}$, at the expense of the $GSC^{low}$, population. At d6, the fraction of the $GSC^{high}$ population was indistinguishable in both treatments relative to the control; however, there was a significant increase in the terminal ADE population ($G^+H^+$; **Fig 6B and 6C**). Therefore, although pre-treatment with EPZ led to an early enhancement of differentiation, both ES_EPZ and ADE_EPZ regimes resulted in elevated ADE production by d6. This suggested that PRC2 inhibition increases the efficiency of endoderm differentiation.

Gene expression analysis on d6 populations prepared under each condition showed that EPZ treatment (ES_EPZ and ADE_EPZ) resulted in differential gene expression relative to DMSO controls; skewed towards gene activation in line with the repressive function of PRC2 (**Fig 6D and 6E**; **S1 Table**). Although the expression profiles in both EPZ treated populations were highly overlapping, ADE_EPZ yielded the most pronounced shift (**Figs 6D and 6E** and **S8A–S8B**). Despite these changes, PCA analysis demonstrated that both EPZ treatment regimens yielded ADE populations that were largely equivalent to their untreated counterparts (**Fig 6F**). Given the apparent discrepancy between this finding and the observed alterations in differentiation efficiency identified by FACS (**Fig 6B and 7C**), we further characterised the genes whose expression was significantly altered in response to PRC2 inhibition. The aggregate gene expression changes that followed EPZ treatment represented an enhancement of that observed between the mesendodermal ($G^{high}$) and ADE ($G^+H^+$) untreated populations (**Figs 6E and S8A–S8B**). This suggested that the ADE gene expression signature was more prominent in EPZ treated cells, consistent with the observed expansion of this population.

Despite enhanced differentiation and a reduced fraction of undifferentiated cells in the ADE_EPZ population, we observed ectopic expression of pluripotency factors in response to PRC2 inhibition (e.g. *Dppa5a* and *Rex1*; **Fig 6F**). We identified genes that showed altered expression upon EPZ treatment but, no equivalent change during ADE differentiation; focussing particularly on those upregulated in EPZ to enrich for direct targets of PRC2. As expected, functional annotation identified enrichment of genes important for embryonic development, however terms associated with inappropriate lineage commitment were also identified (**S8C Fig**). Upregulation of genes involved in neuronal and placental development suggested that, whilst PRC2 inhibition enhanced the transcriptional response to signaling in endoderm

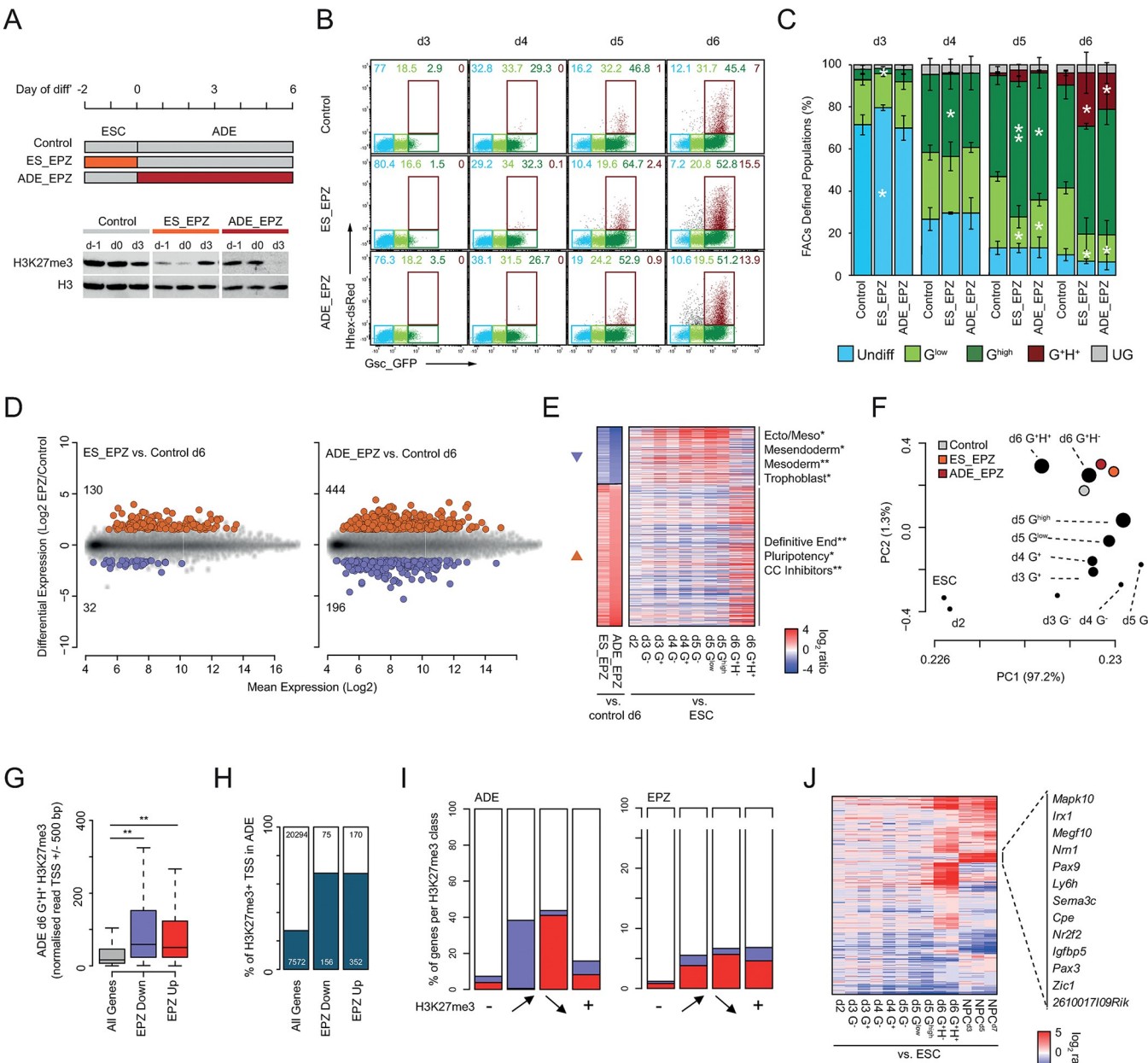

**Fig 6. PRC2 inhibition enhances ADE differentiation at the cost of fidelity. A)** Top; schematic of the EPZ6438 (EPZ) treatment regimen. Below; immunoblot analysis of H3K27me3 and histone H3 for the indicated drug treatments and differentiation samples. **B)** Example FACS profiles comparing GSC-GFP (x-axis) and HHEX-RedStar (y-axis) expression across d3-6 of differentiation treated with DMSO only (Control), EPZ followed by DMSO (ES_EPZ) and DMSO followed by EPZ (ADE_EPZ). The gates used for quantitation are shown and coloured according to reporter (GSC and HHEX) expression as indicated. Data is shown for a single representative replicate experiment (one of three). **C)** Cumulative barplots comparing the relative abundance of each gated population across ADE d3-6 under the indicated treatment regimens. Barplots are scaled to 100% of live FACS sorted cells with error bars representing the standard deviation of three independent replicate differentiations. The significance of differential population size between EPZ treatment (ES_EPZ or ADE_EPZ) and DMSO control were determined for each day and population using a one-sided paired t.test (p≤0.05* and p≤0.01**). **D)** MA plots of differentially expressed genes in ES_EPZ (left) or ADE_EPZ (right) vs control. Upregulated and downregulated genes are coloured in red and blue respectively and the number of differentially expressed genes for each condition are noted. Data represents the mean of three independent replicate experiments. **E)** Heatmap comparing gene expression changes upon EPZ treatment at ADE d6 (log$_2$ fold change of ES_EPZ/Control or ADE_EPZ/Control) and expression changes that occur during untreated ADE differentiation (log$_2$ fold change of ADE$^{Population}$/ESC). Downregulated and upregulated genes are indicated with blue and red arrow heads respectively. Significantly enriched functional gene sets for each of the clusters are indicated adjacent to their respective heatmap (p≤0.05* and p≤0.01** using a Fischer's exact test with Benjamini & Hochberg multiple testing correction). **F)** Scatter plot showing principal components 1 and 2 of PCA analysis performed on the expression data shown in **D** and **E** combined with expression data shown in **Fig 1** for untreated ADE (circle size represents the level of GSC expression based on the FACS gating). The percentage variance attributed to each principal component (PC) is shown in the axis

labels. **G**) Boxplot of TSS H3K27me3 levels (+/- 500 bp) for all, EPZ downregulated and EPZ upregulated genes (d6 ADE EPZ vs. DMSO). The significance of changes in H3K27me3 levels was determined using a Wilcoxon rank-sum test (p≤0.01**). **H**) Cumulative barplots comparing the relative fraction of genes that are associated with an enriched 'peak' of H3K27me3 at their TSS (+/- 500 bp) in one or more ADE populations. Barplots are scaled to 100% of the total genes in each set (total numbers of genes are indicated). **I**). Cumulative barplots comparing the relative fraction of gene subsets classified based on H3K27me3 status (as defined in **S8 Fig**) that are differentially expressed during ADE (left) or upon EPZ treatment (right). Barplots are scaled to 100% of total genes in each subset (n = 20294, 527, 600 and 6445 for '-', gain, lose and '+' categories respectively). **J**). Heatmap comparing gene expression changes during ADE (log2 fold change of ADE$^{population}$/ESC) and NPC differentiations (log$_2$ fold change of NPC$^{day}$/ESC) for the subset of genes with a consistent level of H3K27me3 during ADE and upregulated upon EPZ treatment during ADE differentiation (denoted '+' in **S8 Fig**). Genes that are upregulated in NPCs specifically are indicated to the right.

differentiation, it also allowed for ectopic expression of non-lineage appropriate PRC2 target genes. Indeed, EPZ treatment disproportionately altered the expression of genes that were enriched for H3K27me3 levels at their TSS (**Fig 6G–6H** and **S5 Table**). We classified genes based on whether they lacked ('-'), gained, lost or had constitutive ('+') TSS associated H3K27me3 levels during differentiation (**S8D Fig**). As previously shown, genes that were up- or down-regulated during endoderm differentiation showed a reciprocal loss or gain of H3K27me3 (**Figs 4D** and **6I**; left panel). In contrast, genes that were differentially expressed upon EPZ treatment were enriched in all H3K27me3 categories (**Fig 6I**; right panel). To investigate non-lineage appropriate changes in gene expression, we compared transcription levels during an ESC to neural progenitor cell (NPC) differentiation [51–53] at the subset of genes that were constitutively associated with H3K27me3 and upregulated upon EPZ treatment during ADE differentiation (n = 270 genes; **Fig 6J**). Strikingly, this comparison showed specific up-regulation of a subset of neural genes, including mitogen-activated protein kinase 10 (*Mapk10*), neuritin 1 (*Nrn1*), carboxypeptidase E (*Cpe*) and zinc finger of the cerebellum (*Zic1*), supporting the notion that PRC2 inhibition enhanced ADE differentiation while simultaneously promoting ectopic expression of non-lineage appropriate genes.

## Discussion

Here, we employed a highly reproducible 2D mouse ES differentiation model [30–32] to generate a temporal map of the epigenetic and transcriptional landscape as pluripotent stem cells differentiate through a PS stage toward definitive endoderm. The cell-state transitions that occur during this differentiation model mimics those observed in the gastrulating embryo. We leveraged this system to determine the temporal relationship between histone modification levels and transcription, to assess their contribution to gene regulation, and to determine the impact of PRC2 deficiency on the differentiation pathway.

### H3K27me3 and the dynamics of gene regulation—Cause or Consequence?

Our data show that H3K4me3 tracks with the transcriptional upregulation of genes during differentiation and is not predictive of subsequent up-regulation. This is consistent with studies demonstrating that H3K4me3 deposition and transcription are intimately and reciprocally coupled [54–58]. In contrast, a reduction in H3K27me3 at upregulated genes during the stage equivalent to the epiblast to mesendoderm transition was delayed relative to the up-regulation of transcription by as much as 24 h (**Figs 5** and **S6**). This aligns with our previous observation that small transient transcriptional fluctuations characteristic of ESC Epi-PrEn lineage priming are not accompanied by a detectable shift in promoter proximal H3K27me3 levels [59]. Together, these findings suggest that gene activation does not need the prior removal of H3K27me3, and supports the notion that PcGs are not de facto transcriptional repressors, but rather act to enable fluctuations by preventing its escalation beyond a modest level. This concept mirrors the situation in *Drosophila*, where PcG proteins bind to the regulatory elements

of repressed genes to stabilise and perpetuate transcriptional silencing rather than to initiate it [60–62].

It has been shown that a lack of transcription is a prerequisite for PRC association to CGIs [45,63,64] and that inhibition of transcription leads to PRC2 recruitment to previously 'active' CGIs [65]. These studies suggest that polycomb recruitment to CGIs is the default state, counteracted by transcription, as previously suggested in flies [62] and in mammals [66]. While our findings corroborate the idea that PRCs acts to resist transcription to ensure the appropriate gene expression landscape is established during developmental programmes, our data do not support an instructive role for PRCs in establishing the transcriptional landscape of specific cell types.

## H3K27me3 and the canalization of lineage specific transcription

Loss of function mutations in core mammalian PcG components result in developmental failure between implantation and gastrulation, making it difficult to determine the exact nature of the resulting transcriptional dysregulation that occurs in vivo at such early developmental stages. A recent study has shown that loss of PRC2 catalytic activity phenocopies PRC2 deficiency in mESCs [67] and so acute inhibition is a good proxy for loss of PRC2 function during PS formation. Here we find that inhibition of PRC2, either directly before or during directed endoderm differentiation, resulted in a global reduction in H3K27me3 levels, and increased the production of cells that express both HHEX and GSC and that have a gene expression profile that substantially overlaps with that of ADE (**Fig 6**) suggestive of enhanced ADE differentiation. These findings are however at odds with previous work that found that reduced H3K27me3 deposition caused impaired mesendodermal differentiation [68]. Whilst the reason for this discrepancy is unknown, the conclusions from this study were based on constitutive genetic PRC2 ablation rather than short-term acute PRC2 inhibition. The elevated levels of mesendodermal gene expression observed in PRC2 deficient ESCs could have directly impaired differentiation and/or reduced the relative fold transcriptional activation observed. More likely however, is the fact that this study uses an un-constrained differentiation paradigm based on LIF withdrawal, whereas here we employ a defined regimen of cytokines to direct endodermal differentiation [68].

Although the endoderm expression programme was enhanced by acute PRC2 inhibition, we also found ectopic expression of non-lineage appropriate markers, including pluripotency factors, as has been observed previously [68]. This suggests that loss of PRC2 activity allows for inappropriate erosion of repression that would normally act to safeguard the expression programme during developmental cell state transitions. In *Drosophila*, there is evidence that polycomb buffers against developmental noise [69]. In mammals, loss of either PRC1 or PRC2 leads to perturbed brain development due to the inappropriate upregulation of TFs [70–72]. Similarly, loss of PRC2 specifically in striatal neurons in adult mice leads to the de-repression of an ectopic gene expression programme and neurodegeneration [73].

Conrad Waddington introduced the concept of canalization as a buffer against genetic and environmental variability perturbing the phenotype [33]. Cells progress down the valleys blocked from lineage inappropriate decisions and the escape from lineage trajectories. The hills and troughs that funnel cell state transitions and developmental decisions are sculpted by a combination of the signalling environment, gene regulatory networks and epigenetic mechanisms that modulate transcription. Our findings suggest that PRC factors both restrain exit from the desired trajectory by canalizing gene expression programmes and control the incline upon which a cell tracks towards a new state.

## Materials and methods

### ESC culture, ADE differentiation and EPZ treatment

HRS/Gsc-GFP dual reporter cells ([74]; **S1A Fig**) were cultured in ESC medium + LIF as previously described [59] and differentiated according to our standard ADE differentiation protocol [32]. To generate sufficient cells for Fluorescence Activated Cell Sorting (FACS), ADE differentiation was performed in T75 flasks (Corning, Corning, NY 14831) alongside a 6-well plate (Corning) that was analysed by flow cytometry at d6 to ensure differentiation occurred as expected. As cell density increased up to day 5 of differentiation, and the populations to be isolated constituted different relative proportions of the total cell number for each day (**S1C Fig**), the number of T75 flasks required at different time points varied. For inhibitor treatments, cultures were supplemented with either 1:4000 DMSO (control) or 2.5 μM EPZ6438 (EPZ; from 10 mM stock in DMSO) as indicated in **Fig 6**.

### Quantitative RT-PCR

SuperScript III reverse transcriptase (Thermo Fisher Scientific) was used to perform first-strand synthesis on 1 μg of total RNA according to the manufacturer's instructions. Standard curves were generated using a mix of concentrated cDNA from all samples and cDNA equivalent to 20 ng of total RNA was used in each reaction. Amplification was detected using the Universal Probe Library system on a Roche LightCycler 480 (Roche, Germany) and data was normalised to the geometric mean of housekeeping genes *Tbp*, *Pgk1* and *Sdha* and analysed using the methods described in [75]. Raw qRT-PCR data is provided in **S8 Table**.

### Flow cytometry and analysis

For flow cytometry analysis, HRS/Gsc-GFP dual reporter cells at day 6 of the ADE differentiation were isolated by washing the cells twice with PBS (Sigma; suitable for cell culture; without CaCl or MgCl) and after aspiration adding 1 ml/well attenuated trypsin (0.025% trypsin (Invitrogen 15090–046), 1.27 mM EDTA, 1% Chicken serum (Sigma C5405) in PBS) and incubating at 37˚C for 2–3 mins. The plate was tapped vigorously and trypsin was then inactivated by adding 9 ml FC buffer (10% fetal calf serum in PBS), followed by extensive pipetting to obtain a single cell suspension. The cells were counted using a Scepter cell counter or a Neubauer hemocytometer and kept on ice during the antibody staining. All centrifugation was performed at 330 g for 2–3 mins. Cells were washed twice by centrifuging, aspirating, adding 10 ml FC buffer and pipetting to re-establish single cell suspension. Antibody staining for Cxcr4 was performed for 20 mins on ice using 1:100 APC-conjugated Rat anti-Mouse CD184/Cxcr4 (BD Pharmingen) in FC buffer and cells were then washed twice in FC buffer and resuspended in 1 μg/ml DAPI in FC buffer at ~10–15 M cells/ml to label dead cells and run the suspension through a cell strainer. Flow cytometry analysis was performed using a BD LSR Fortessa. Visualisation and further analysis was performed using FCS Express 4 Flow Research. FACS data used for plotting and statistical analysis is provided in **S8 Table**.

### Microscopy

To generate the images shown in **S1B Fig**, HRS/Gsc-GFP dual reporter cells were plated on 8-well μ-Slides (Ibidi, cat. no. 80821) coated with 0.1% gelatin and differentiated according to our previously described ADE differentiation protocol [32]. After washing with PBS, the cells were imaged at 100x magnification using a Nikon widefield fluorescence microscope. Imaging settings were stored and reapplied to acquire images on consecutive days. The resulting images

were subjected to rolling ball background subtraction and contrast brightness enhancement in ImageJ, applied equally across all images.

## RNA extraction and microarray processing

RNA was extracted from $0.1–0.5 \times 10^6$ ESCs / FACS purified ADE cells using an RNeasy Mini kit (Qiagen; 74104). RNA quantification and assessment of integrity was performed using the total RNA nano chip (Agilent; 5067–1511) and measured on an Agilent Bioanalyzer. 150 ng of purified RNA was mixed with RNA standards (One Colour RNA Spike-In Kit; Agilent 5188–5282) and the mixture labelled with LowInput QuickAmp Labeling Kit One-Color (Agilent 5190–2305). For each sample, 600 ng of Cy3-labelled cRNA was fragmented at 60˚C for 30 min then mixed with an equal volume of 2x Agilent hybridization buffer and hybridised to an Agilent-028005 SurePrint G3 Mouse GE 8x60K Microarray (Agilent; G4852A) for 17 h at 65˚C. Following hybridisations, microarrays were washed and scanned on a Surescan G2600D scanner (serial: SG12524268) using Agilent Scan Control 9.1.7.1 software with default settings. Signal intensity values were extracted from the resulting single channel TIFF images using Agilent feature extraction software (Agilent Feature Extraction 11.0.1.1) with default parameters (AgilentG3_HiSen_GX_1color). All steps were carried out in accordance with manufacturer's instructions unless otherwise stated.

## 4sU sample preparation and sequencing

4sU incubation was performed in ADE differentiation medium (ADEM) from the T75 flasks to be used for sorting. To obtain the ADEM, 15 ml of ADEM was pipetted from each T75 flask, flasks were returned to the incubator and the extracted ADEM was pooled and centrifuged at 1000 g for 5 mins to remove cell debris. In the following steps, flasks were treated in the same order for every action taken in batches of up to 6 flasks to avoid timing differences between flasks. After aspiration, 5 ml pooled ADEM (negative control; 1–2 flasks per sort) or 5 ml pooled ADEM containing 500 µM 4sU were added and flasks were then returned to the incubator. After exactly 20 mins, flasks were aspirated, washed with 10 ml PBS (Merck; D8537), aspirated again and incubated for 3 min at 37˚C in 2ml per flask of attenuated trypsin (0.025% (v/v) trypsin (Thermo Fisher Scientific; 15090–046), 1.27 mM EDTA, and 1% (v/v) Chicken serum (Merck; C5405) in PBS (Merck; D8537)). Cells were then released from the flask by tapping and attenuated trypsin was inactivated by adding 8 ml ice-cold flow cytometry buffer (FC buffer—10% FCS in PBS). Cell suspensions from all flasks with the same treatment in the batch were then pooled, pipetted repeatedly to obtain a single cell suspension, counted using a hemocytometer (Neubauer) and stored on ice. This process was repeated when required (for d3 & d4 samples) until a sufficient number of cells were obtained for the sorting procedure ($\sim 1 \times 10^8$ cells). All centrifugation steps hereafter were performed at 400 g for 3 mins. Samples were centrifuged, aspirated, resuspended in ice-cold FC buffer at $1–1.5 \times 10^7$ cells/ml and passed through a cell strainer (Corning; 352235). Negative control cells were not sorted and kept on ice during the sorting procedure. 4sU-labelled cells were sorted on a BD FACSAriaIII into 5 ml Falcon tubes (Corning) at 4˚C containing 1 ml of ice-cold FC buffer. After sorting 1 million cells, the tube was transferred to ice in the dark and replaced with a fresh 5 ml Falcon tube containing 1 ml of ice-cold FC buffer and this process was repeated until all 4sU-labelled cells were sorted. Samples were centrifuged, resuspended in 100 µl of ice-cold PBS and cells from the same population were pooled to obtain an equal number of cells for each population, equivalent to the population that generated the lowest number of cells and an identical number of cells was taken from the unsorted and unlabelled negative control sample. All samples were then transferred to 1.5 ml DNA LoBind tubes (Eppendorf), added up

to 1 ml using ice-cold PBS, centrifuged, aspirated and agitated to resuspend cells in the remaining PBS (<10 µl). Cells were lysed by adding 1 ml of Trizol reagent (Thermo Fisher Scientific) equilibrated to RT. After 5 mins of incubation at RT, samples were stored at -20˚C. 4sU incorporated RNAs were isolated as previously described [3]. Individual purified libraries were pooled into two six-sample equimolar pools containing the indexes 3–8 (NEBNext multiplex oligos for Illumina, set 1, E7335) and deep-sequenced at BGI on a HiSeq4000 (paired-end 100-bp reads).

## Native chromatin immunoprecipitation and sequencing

Native ChIP was performed on $0.5-1 \times 10^6$ flow sorted ADE cells and immunopurified with either anti-H3K4me3 (07–473; Millipore) or anti-H3K27me3 (07–449; Millipore) as previously described [59]. ChIP libraries were prepared according to [56] with modifications described in [59]. Replicate sample libraries (n = 2/3) were either combined into equimolar pools prior to sequencing or in silico following sequencing. Libraries were sequenced on an Illumina Hi-seq 2000 system following the standard Illumina protocol to generate single-end 42 or 49 bp reads (The Danish National High-Throughput DNA Sequencing Centre).

## Computational analysis

**Microarray processing and gene expression analysis.**   Microarray data was processed with custom scripts in R. Briefly, raw signal quantitation files were read using the 'read.maimages' function in the limma package in R with source set to "agilent" and annotation based on the Agilent annotation file ('028005_D_20130207.gal') [76]. Duplicate probe intensity values were replaced with their arithmetic mean. Processed intensity values were produced using the limma package by background correction ('normexp' with an offset of 10) followed by quantile normalisation across all samples (R/Bioconductor; [76,77] (http://www.R-project.org/ )). Differential gene expression analysis was performed on $\log_2$ normalised intensity values for all 55,681 unique probes (excluding control probes) with linear modelling and eBaysean statistics implemented using the limma 'topTable' function with Benjamini-Hochberg multiple testing correction for each sample comparison. Genes were considered to be differentially expressed if they had a corrected p value of $\leq 0.01$ and a $\log_2$ fold change (FC) of $\geq 1.5$ or $\leq$ -1.5 (or a cut-off p value of $\leq 0.05$ in the mRNNA vs 4sU analysis in **Fig 2**).

**Native ChIP-seq mapping, processing and visualisation.**   For each sample (ADE D3-6 ChIP-seq and ADE Day 6 inputs), multiple raw Fastq files were merged together and mapped to the mouse genome (mm9) using bowtie2 with the following options (—local -D 20 -R 3 -N 1 -L 20 -i S,1,0.50) [78]. Using the HOMER package [79], SAM files were converted into tag directories and multi-mapping reads were removed using makeTagDirectory (parameters: -unique -fragLength 150). Mapped regions that, due to fragment processing, extended beyond the end of the chromosomes were removed using removeOutOfBoundsReads.pl. Peak finding was performed using the MACS2 (v2.1.1) 'callpeak' function with default settings [80]. Peaks with a MACS2 threshold of $\geq$25 were retained and combined into a single interval file for either H3K27me3 or H3K4me3 using the BEDTOOLs 'mergeBed' function with the following parameters (-d 300) [81]. Genes were considered to be associated with an H3K27me3 peak if they were within ± 500 bp of a Refseq TSS determined using the BEDTOOLs 'intersect' function [81]. The same approach was used to intersect gene TSSs with biochemically annotated CGIs [82]. For visualisation, genome browser files (.bw) were generated using makeUCSCfile with the following options (-bigWig -fsize 1e20) with normalisation (-norm) set to $4 \times 10^6$ and $10 \times 10^6$ reads for H3K4me3 and H3K27me3 respectively. Signal quantitation was performed for Refseq annotated TSSs using the HOMER package. For region quantitation, read coverage

was determined at TSS (+/- 500 bp) with the following parameters (-size "given" -noann -nogene -noadj). For heatmaps, read coverage was determined (+/- 5 kb) with the following parameters (-size "given" -noann -nogene -noadj -hist 100 -ghist). Data visualisations and statistical analysis were performed using custom R scripts.

**4SU-seq mapping, processing and visualisation.** For mapping and manipulation of 4sU-seq data from the ADE differentiation generated in this study and published ESC to NPC differentiation data (GSE115774 [53]). Following demultiplexing, Fastq files were aligned to the mouse genome (mm9 build) using Bowtie2 v2.2.6 for paired-end sequence data with default settings. Aligned.SAM files were converted into tag directories using the HOMER package (v4.8) using the "makeTagDirectory" function with the following parameters (-format sam -flip–sspe) [79]. Genomic intervals that extended beyond the end of the chromosomes were removed using the "removeOutOfBoundsReads.pl" function. Strand-specific browser track files in "bigWig" format were generated and combined replicate data using "makeUCSCfile" with the following parameters (-fsize 1e20 -strand +/− -norm 5e7 -lastTag -color 25,50,200). HOMER was used to quantify 4sU-seq read coverage across all gene introns and/or exons based on a Refseq annotation (mm9 genome build; '.gtf' format). Coverage was determined using 'analyzeRepeats.pl' with the following parameters (-count exons/introns -strand + -noadj). For ADE, expression values were converted into reads per kilobase (RPK) and differential gene expression analysis performed as for the microarray-based gene expression analysis (using custom scripts in R). For published NPC data (GSE115774 [53]), values were averaged from two independent replicates, converted into RPK values (plus a one read offset) and then normalised across ESCs and NPCs using 'normalizeQuantiles' from the limma package (R/Bioconductor; [76,77] (http://www.R-project.org/)).

For the dReg analysis of enhancers, 4sU-seq data in BAM file format was run through the dReg algorithm via the web-based interface with default parameters (https://dreg.dnasequence.org/; [41]). dReg peaks were combined using the bedtools 'merge' function and then read signal was quantified across these regions using the 'annotatePeaks.pl' in the HOMER package using the following parameters (-size given -noann -nogene -norm 100e6 -strand both). Quantitation files to generate heatmaps and meta-profiles were generated using the 'annotatePeaks.pl' in the HOMER with the following parameters (-size 10000 -hist 100 -ghist -noann -nogene -norm 1 x $10^8$ -strand both/-/+). Data visualisations and statistical analysis were performed using custom R scripts.

**Cluster analysis.** All clustering was performed on distance matrices computed using the 'dist' function in R (method = euclidian). The resulting matrices were clustered using the 'hclust' function in R (method = complete). For microarray expression data, clustering was performed on correlation scores for the expression profiles of each gene across the differentiation (**Fig 1**). For ChIP-seq data, correlations were computed for all TSSs (+/- 500 bp) based on the d3 G⁻ and d6 G⁺H⁺ H3K27me3 and H3K4me3 data. For this analysis, ChIP quantification data was scaled (using the 'scale' function in R) to ensure an equal contribution from both modifications to the clustering outcome (R/Bioconductor; [77] (http://www.R-project.org/)).

**Gene function analysis.** Functional enrichment analysis (Gene Ontology; GO) was performed using the G Profiler algorithm for both 'Cell Component' and 'Biological Process' terms. Results were filtered based on an FDR of 0.01. Functional enrichment analysis for genes grouped based on clustering of TSS ChIP-seq signal (**S5 Fig**) was performed using the 'enrichGO' function in the clusterProfiler R package with the following parameters (OrgDb = "org.Mm.eg.db", keyType = "SYMBOL", ont = "BP", pvalueCutoff = 0.01 and pAdjustMethod = "BH") and visualised using the ggplot2 package using custom scripts [83]. The full list of results are provided (**S6 Table**). For targeted analysis, enrichment of manually annotated gene sets (**S2 Table**) was determined by comparing the intersection between functional lists and

differentially expressed lists. Statistical analysis was performed using a Fishers test and Benjamini–Hochberg correction for multiple testing.

## Supporting information

**S1 Fig. Validation of in vitro differentiation.** A) Targeting strategies employed for the construction of the B6 GSC-GFP and HHEX-RedStar double reporter mESC line [31,35]. B) Fluorescent and brightfield images (upper panel) and FACS scatter plot profiles (lower panel) of B6 reporter ESCs undergoing ADE differentiation on days 3–6. FACS profiles include GSC-GFP and HHEX-RedStar reporter fluorescence and immunofluorescent detection of the definitive mesendoderm marker CXCR4 (using anti-CXCR4-APC). Scale bars; 100 μm. C) FACs analysis profiles to determine sample drift between the indicated gates following the FACS collection period ($\leq$ 2 h). Percentages in the original gates and in the post sort analysis are indicated. D) Quantitative RT PCR analysis of *Gsc*, *Hhex*, *Pou5f1* and *Cer1*. Relative expression between differentiation samples was calculated by normalizing the transcript number by the geometric mean of *Tbp*, *Pgk1* and *Sdha* expression. Error bars represent the standard deviation of the mean of $\geq$ 3 independent experiments.
(PDF)

**S2 Fig. Gene expression changes during in vitro differentiation. A**) Pairwise scatter plots (left panel) and summary barplot (right panel) showing the number of differentially expressed genes between each ADE population vs. ESCs. Upregulated and down regulated genes are shown in red and blue respectively. The number of significant genes are noted in parenthesis (RefSeq annotation). Differentially expressed genes are defined as those with a $\log_2$ fold change of $\geq$ 1.5 and an adjusted p value of $\leq$ 0.01 (Benjamini & Hochberg multiple testing correction). **B**) Plots depicting the aggregate expression profile of the genes in each of the clusters defined in **Fig 1E**. The heavy dashed line, dark grey shaded area and light grey shaded areas represent the median, $25^{th}$ to $75^{th}$ percentile range and $10^{th}$ to $90^{th}$ percentile range of the log2 fold change for each gene set respectively. Significantly enriched functional gene sets for each of the clusters are indicated below their respective plot ($p \leq 0.05^*$ and $p \leq 0.01^{**}$ using a Fischer's exact test with Benjamini & Hochberg multiple testing correction). Significantly enriched transcription factor motifs proximal to the TSSs (+100 bp to -50bp) are indicated to the right of each plot ($p \leq 0.01$ Benjamini multiple testing correction).
(PDF)

**S3 Fig. Genes upregulated in 4sU-seq only have a higher proportion of unspliced transcripts. A**) Boxplots showing length corrected $\log_2$ exon/intron ratios for all (grey) or upregulated (red; in d3 G$^+$ vs. G$^-$) for the indicated populations. Significant changes in $\log_2$ ratios were determined using a Wilcoxon rank-sum test (p values as displayed). Data represents the mean of three independent replicate experiments.
(PDF)

**S4 Fig. Transcriptional profiling identifies prospective enhancer signatures at loci proximal to differentially regulated genes. A**) Genome browser tracks of normalised 4sU-seq signal at candidate gene loci with differentially expressed dREG peaks proximal to differentially regulated genes. Data and genes are presented as in **Fig 2A** and dREG peaks are indicated with black arrow heads. **B**) Boxplots showing the $\log_2$ ratio (G$^+$/G$^-$) of normalised 4sU-seq signal for dREG peaks across a range of distance separations from differentially regulated genes (in d3 G$^+$ vs. G$^-$). Plots show the $\log_2$ ratios for d3 and d4 of differentiation (upper and lower panels respectively) at dREG peaks associated with downregulated (blue) and upregulated genes (red). Boxplots depict the median and 25%-75% data distribution (black bar and coloured

box respectively). Significant changes between the populations for each distance and condition for the raw, unlogged values, are indicated above their respective plot (p≤0.05* and p≤0.01** determined using a Wilcoxon signed rank test). Data shown in (**A**) and (**B**) represent the mean of three independent replicate experiments.
(PDF)

**S5 Fig. Dynamic changes in H3K4me3 and H3K27me3 during differentiation.** Genome browser tracks of ChIP-seq signal for H3K4me3 (left panel) and H3K27me3 (right panel) at the **A**) *Gapdh* and **B**) *Hoxb* gene loci (annotated as per the mm9 genome assembly). **C**) Heat-maps of H3K4me3 (left) and H3K27me3 (right) ChIP-seq signal spanning +/- 5 kb of gene TSSs grouped and ordered based on clustering of the TSS (+/- 500 bp) ChIP-seq signal from d3 $G^-$ and d6 $G^+H^+$ populations for both histone modifications. Heatmaps are sub-divided into 11 clustered groups, and the percentage of genes with a CGI TSS in each group is shown in parenthesis. The signal scale for each modification is shown below their respective heatmap. **D**) Summary boxplots of normalised H3K4me3 and H3K27me3 ChIP-seq signal at the TSSs (+/- 500 bp; left and middle panel respectively) and gene expression levels (right panel) for six representative clusters from panel (**D**). **E**) The top five enriched functional gene ontology terms (biological process) for each of the ChIP-defined clusters shown in (**C**; N.B. cluster seven is absent as it lacked any significantly enriched terms). Gene count and adjusted p-values are indicated as per the key. Full lists of enriched functional terms for each cluster are provided in **S6 Table**. ChIP-seq data shown in (**A**—**D**) and gene expression data shown in (**D**) represent the mean of two and three independent replicate experiments respectively.
(PDF)

**S6 Fig. Changes in TSS H3K27me3 levels are temporally subordinate to changes in gene transcription.** **A**) MA plots of gene sets that show consistent differential transcription/4sU-seq signal between the $G^+$ and $G^-$ populations at d3 and d4 of ADE differentiation. Upregu-lated and downregulated genes are coloured in red and blue respectively and the number of genes are noted in parenthesis in the lower panel. **B**) Summary boxplots of $\log_2 G^+/G^-$ ratios for the gene and datasets shown in panel **A**. **C**) Scatter plots depicting the $\log_2 G^+/G^-$ transcrip-tion/4sU-seq signal ratios (x axis) vs. the $\log_2 G^+/G^-$ ratio of H3K4me3 ChIP-seq signal at the TSS (+/- 500bp; y axis) of the differentially transcribed genes shown in **A**. **D**) Summary box-plots of the $\log_2 G^+/G^-$ ratio of H3K4me3 ChIP-seq signal at the TSS (+/- 500 bp; y axis) for the gene and datasets shown in (**C**). Significant changes between the populations for each day for the raw, unlogged values, are indicated above their respective plot (p≤0.05* and p≤0.01** determined using a Wilcoxon signed rank test). **E**) As for (**C**) but for H3K27me3 ChIP-seq data. **F**) As for (**D**) but for H3K27me3 ChIP-seq data. 4sU-seq data (**A** and **B**) and ChIP-seq data (**C**—**F**) represent the mean of three and two independent replicate experiments respec-tively.
(PDF)

**S7 Fig. Invariant TSS H3K27me3 at genes with high levels of transcriptional activation in $G^+$ vs. $G^-$ cell populations. A**) Genome browser tracks of ChIP-seq signal for H3K27me3 (upper panel) and 4sU-seq (lower panel) from the d3 $G^+$ population at the *Gsc* gene loci (anno-tated as per the mm9 genome assembly). The 4sU-seq signal is coloured according to the tran-scribed strand (positive—black and negative—grey). Normalised read depths are indicated in parenthesis for each data type. Tracks represent a single matched experiment or combined replicates (n = 2) for 4sU-seq and ChIP-seq respectively. **B**) Scatter plots of all gene TSSs that are upregulated in $G^+$ vs $G^-$ populations for both D3 and D4 of differentiation (Up; in red) and the subset of TSSs that are in the upper 50% of H3K27me3 ChIP-seq and 4sU-seq signal (4sU

+K27+Up; red with black circles) from the d3 G$^+$ population. Plots represent (in order from top to bottom) the H3K27m3 ChIP-seq signal (TSS +/- 500 bp from the d3 G$^+$ population), 4sU-seq signal (from the d3 G$^+$ population) and the ratio of H3K27me3 ChIP-seq signal (TSS +/- 500 bp) between G$^+$ and G$^-$ populations at day 3 and 4 of differentiation. 4sU-seq data and ChIP-seq data represent the mean of three and two independent replicate experiments respectively. **C**) Boxplot of log2 H3K27me3 ratios (G$^+$/G$^-$) for all upregulated and 4sU+K27+ upregulated gene sets at day 3 and day 4 of differentiation (number of genes based on gene symbol annotation). Significant changes between the populations for each gene set are indicated above their respective plot ($> 0.05^{n/s}$, $p \leq 0.05^*$ and $p \leq 0.01^{**}$; determined using a Wilcoxon signed rank test). ChIP-seq data represent the mean of two independent replicate experiments. **D**) Genome browser tracks of ChIP-seq signal for H3K27me3 (upper panel) and 4sU-seq (lower panel) for the genes indicated in panel (**B**; 1–5). Tracks are displayed as in panel (**A**). (PDF)

**S8 Fig. Characterisation of endodermal differentiation following PRC2 inhibition. A** & **B**) Venn diagram comparing the overlap between differentially expressed gene (DEG) sets. Overlap for up and down-regulated DEGs are shown for; ADE d6 G$^+$H$^+$ vs ESCs, ADE d6 G$^+$H$^-$ vs ESCs, ES_EPZ treated ADE d6 vs control treated ADE d6, and ADE_EPZ treated ADE d6 vs control treated ADE d6. **C**) The top 20 significantly enriched biological processes terms from a GProfiler GO analysis performed on the genes set that were upregulated upon EPZ treatment but not in ADE d6 vs ESCs (all displayed results have an adjusted p.value < 0.05). **D**) Heatmap of H3K27me3 ChIP-seq signal at TSSs (+/- 5 kb) subdivided into genes with no/low H3K27me3 levels ('-'), gain H3K27me3 levels, lose H3K27me3 levels or consistent positive H3K27me3 levels ('+') throughout the ADE differentiation for the indicated samples. (PDF)

**S1 Table. Summary Gene Expression Data.** Data table of normalised microarray expression data, differential gene expression analysis statistics and ADE cluster assignment. This includes expression data for ESCs, ADE populations and EPZ treated d6 ADE populations. Data is annotated for both microarray probes and genes (RefSeq and Gene Symbols) for the mm9 genome build and includes gene cluster assignments as defined in **Fig 1**. (XLSX)

**S2 Table. Marker Gene Lists. Data table listing the marker gene sets used in enrichment analyses.** (XLSX)

**S3 Table. Expression Data for Day 3 and 4 ADE Populations.** Data table of normalised expression data (total RNA/microarray and 4sU-seq) and differential gene expression analysis statistics for ADE d3 and d4 G$^+$ and G$^-$ populations. Data is annotated for both microarray probes and genes (RefSeq and Gene Symbols) for the mm9 genome build and includes gene set assignments as defined in **Fig 2**. 4sU-seq expression data is given as normalised read per kilobase (RPK) values for both introns and exons (summed values for all introns and exons per gene where applicable). (XLSX)

**S4 Table. 4sU-seq Signal at Putative Enhancers in Day 3 and 4 ADE Populations.** Data table of normalised 4sU-seq signal at putative enhancer 'peaks' identified in Day 3 and 4 ADE populations by the dREG algorithm. Regions are annotated according to the mm9 genome assembly. Proximity of dREG peaks to the nearest neighbouring gene (RefSeq annotation) within the specified size windows is indicated (0 and 1 indicate no-overlap and overlap within

the specified window respectively).
(XLSX)

**S5 Table. Summary H3K4me3 and H3K27me3 ChIP-seq Data.** Data table of normalised expression data (total RNA/microarray and 4sU-seq), differential gene expression statistics (total RNA/microarray and 4sU-seq) and ChIP-seq signal (H3K4me3 and H3K27me3) for all mouse genes across the ADE populations (annotated with microarray probe ID, RefSeq and gene symbol annotations). Normalised signal for ChIP-seq and input samples are given for RefSeq gene TSSs (+/- 500 bp; mm9 genome build coordinates). Differential expression cluster assignment (relating to **Figs 5 and S6**) is indicated by '0' and '1'.
(XLSX)

**S6 Table. Functional Enrichment Analysis for Histone-Mark-Defined Gene Clusters.** Summary of significantly enriched functional terms (biological process) for gene sets clustered based on TSS-associated (+/- 500 bp; mm9) histone mark signal (H3K4me3 and H3K27me3) across the ADE differentiation. Gene set clusters defined as indicated in **S5C–S5F Fig**.
(XLSX)

**S7 Table. Summary Gene Expression Data Comparing ADE and NPC Differentiations.** Data table of differential gene expression analysis statistics for ADE and NPC differentiations (relating to **Fig 6**). ADE expression data was taken from this study and compared to NPC differentiation data from [53].
(XLSX)

**S8 Table. Raw Data Underlying Figs 6C and S1D.** Raw data values used to compute plots shown in **Figs 6C** and **S1D.** Specific details are indicated for each dataset within the respective tables.
(XLSX)

## Acknowledgments

We would like to thank Elisabeth Freyer (IGC) and Gelo Dela Cruz for FACS and the Danish National High-Throughput DNA Sequencing Centre for next generation sequencing. We are grateful to members of the Illingworth, Brickman and Bickmore labs for critical comments on this study

## Author Contributions

**Conceptualization:** Jurriaan Jochem Hölzenspies, Wendy Anne Bickmore, Joshua Mark Brickman, Robert Scott Illingworth.

**Data curation:** Jurriaan Jochem Hölzenspies, Dipta Sengupta, Wendy Anne Bickmore, Joshua Mark Brickman, Robert Scott Illingworth.

**Formal analysis:** Jurriaan Jochem Hölzenspies, Dipta Sengupta, Robert Scott Illingworth.

**Funding acquisition:** Dipta Sengupta, Wendy Anne Bickmore, Joshua Mark Brickman, Robert Scott Illingworth.

**Investigation:** Jurriaan Jochem Hölzenspies, Dipta Sengupta, Robert Scott Illingworth.

**Methodology:** Jurriaan Jochem Hölzenspies, Joshua Mark Brickman, Robert Scott Illingworth.

**Project administration:** Jurriaan Jochem Hölzenspies, Joshua Mark Brickman, Robert Scott Illingworth.

**Resources:** Wendy Anne Bickmore, Joshua Mark Brickman.

**Supervision:** Wendy Anne Bickmore, Joshua Mark Brickman, Robert Scott Illingworth.

**Validation:** Jurriaan Jochem Hölzenspies, Robert Scott Illingworth.

**Visualization:** Robert Scott Illingworth.

**Writing – original draft:** Wendy Anne Bickmore, Joshua Mark Brickman, Robert Scott Illingworth.

**Writing – review & editing:** Jurriaan Jochem Hölzenspies, Wendy Anne Bickmore, Joshua Mark Brickman, Robert Scott Illingworth.

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
