## [Editor Report · Decision Letter 0]

10 Sep 2024

Dear Dr Illingworth,

Thank you very much for submitting your Research Article entitled 'PRC2 Promotes Canalisation During Endodermal Differentiation' to PLOS Genetics.

The manuscript was fully evaluated at the editorial level alongside the Review Commons reviews. The reviewers appreciated the attention to an important problem, but raised some substantial concerns about the current manuscript that you have proposed to address. Thank you for the detailed response to reviewers and proposed research plan for revisions. Please note that the most critical issue is the ability to retain the claim that H3K27me3 is lost after transcriptional activation, and so the detailed analyses and discussion of this point is most critical. We would be willing to review a much-revised version. We cannot, of course, promise publication at that time.

If you decide to revise the manuscript for further consideration at PLOS Genetics, please aim to resubmit within the next 60 days, unless it will take extra time to address the concerns of the reviewers, in which case we would appreciate an expected resubmission date by email to plosgenetics@plos.org.

To resubmit, log into your Editorial Manager account and select the option 'Revise Submission' in the 'Submissions Needing Revision' folder.

We are sorry that we cannot be more positive about your manuscript at this stage. Please do not hesitate to contact us if you have any concerns or questions.

Yours sincerely,

Marnie E. Blewitt

Academic Editor

PLOS Genetics

John Greally

Section Editor

PLOS Genetics

---

## [Decision Letter · Decision Letter 1]

14 Jan 2025

PGENETICS-D-24-00986R1

PRC2 Promotes Canalisation During Endodermal Differentiation

PLOS Genetics

Dear Dr. Illingworth,

Thank you for submitting your manuscript to PLOS Genetics. After careful consideration, we feel that it has merit but does not fully meet PLOS Genetics's publication criteria as it currently stands. Therefore, we invite you to submit a revised version of the manuscript that addresses the points raised during the review process.

Please submit your revised manuscript within 30 days Feb 13 2025 11:59PM. If you will need more time than this to complete your revisions, please reply to this message or contact the journal office at plosgenetics@plos.org. Please include the following items when submitting your revised manuscript:

We look forward to receiving your revised manuscript.

Kind regards,

John M. Greally, D.Med., Ph.D.

Section Editor

PLOS Genetics

John Greally

Section Editor

PLOS Genetics

Aimée Dudley

Editor-in-Chief

PLOS Genetics

Anne Goriely

Editor-in-Chief

PLOS Genetics

**Additional Editor Comments :**

Please address the issue raised by Reviewer 1, that you "should not claim that gene activation does not require the loss of H3K27me3 at the gene promoter (TSS)." If this edit is made, we will be happy to accept the manuscript without further edits.

**Journal Requirements:**

1) When completing the data availability statement of the submission form, you indicated that you will make your data available on acceptance. We strongly recommend all authors decide on a data sharing plan before acceptance, as the process can be lengthy and hold up publication timelines. Please note that, though access restrictions are acceptable now, your entire data will need to be made freely accessible if your manuscript is accepted for publication. This policy applies to all data except where public deposition would breach compliance with the protocol approved by your research ethics board. If you are unable to adhere to our open data policy, please kindly revise your statement to explain your reasoning and we will seek the editor's input on an exemption. Please be assured that, once you have provided your new statement, the assessment of your exemption will not hold up the peer review process.

2) Please include the affiliation of Robert Scott Illingworth in the online submission form.

**Reviewers' comments:**

Reviewer's Responses to Questions

Reviewer #1: As Reviewer 1, I've addressed the response to my previous comments here. Since the authors acknowledge that their current cell-population-based data does not allow them to test whether gene activation in a small subset of cells accounts for the observed results, they should not claim that gene activation does not require the loss of H3K27me3 at the gene promoter (TSS). The data in Figure S7 does not change my opinion on this. It should be fine to instead revise the text to avoid this issue.

Reviewer #2: The authors have replied to all my comments in a very satisfactory manner, I acknowledge them for their efforts. I find the paper ready for publication.

**Have all data underlying the figures and results presented in the manuscript been provided?**

Reviewer #1: Yes

Reviewer #2: None

PLOS authors have the option to publish the peer review history of their article (what does this mean?). If published, this will include your full peer review and any attached files.

Reviewer #1: No

Reviewer #2: No

**Figure resubmission:**
---

## [Editor Report · Decision Letter 2]

20 Jan 2025

Dear Dr Illingworth,

We are pleased to inform you that your manuscript entitled "PRC2 Promotes Canalisation During Endodermal Differentiation" has been editorially accepted for publication in PLOS Genetics. Congratulations!

Yours sincerely,

John M. Greally, D.Med., Ph.D.

Section Editor

PLOS Genetics

John Greally

Section Editor

PLOS Genetics

Aimée Dudley

Editor-in-Chief

PLOS Genetics

Anne Goriely

Editor-in-Chief

PLOS Genetics

Comments from the reviewers (if applicable):

**Data Deposition**

http://datadryad.org/submit?journalID=pgenetics&manu=PGENETICS-D-24-00986R2

**Press Queries**

---

## [Editor Report · Acceptance letter]

25 Jan 2025

PGENETICS-D-24-00986R2 

PRC2 Promotes Canalisation During Endodermal Differentiation 

Dear Dr Illingworth, 

We are pleased to inform you that your manuscript entitled "PRC2 Promotes Canalisation During Endodermal Differentiation" has been formally accepted for publication in PLOS Genetics! Your manuscript is now with our production department and you will be notified of the publication date in due course.

With kind regards,

Zsofia Freund

PLOS Genetics

On behalf of:
